# Semi-Synthetic Localization Datasets for Radiological Findings on Chest X-Rays

**Andrea Posada**[1,2] (iD)                                    ANDREA.POSADA-CARDENAS@TUM.DE
**Johannes Brandt**[1]                                        JOHANNES.BRANDT@TUM.DE
**Friederike Jungmann**[1,3]                                  FRIEDERIKE.JUNGMANN@TUM.DE
**Maria Posada**[4]                                           MARIA.POSADAC@UDEA.EDU.CO
**Daniel Rueckert**[1,2,5]                                    DANIEL.RUECKERT@TUM.DE
**Martin J. Menten**[1,2]                                     MARTIN.MENTEN@TUM.DE
**Felix Meissen**[1]                                          FELIX.MEISSEN@TUM.DE
**Philip Müller**[1]                                          PHILIP.J.MUELLER@TUM.DE

[1] *Chair for AI in Healthcare and Medicine, Technical University of Munich (TUM) and TUM University Hospital, Munich, Germany*

[2] *Munich Center for Machine Learning (MCML), Munich, Germany*

[3] *Department of Diagnostic and Interventional Radiology, TUM University Hospital, Technical University of Munich (TUM), Munich, Germany*

[4] *Independent Researcher, Medellin, Colombia*

[5] *Department of Computing, Imperial College London, UK*

**Editors:** Accepted for publication at MIDL 2026

## Abstract

While large datasets for chest X-ray (CXR) finding classification are widely available, datasets for finding localization are scarce. Curating these localization datasets is costly and time-intensive, requiring manual annotation by medical experts, which often results in them being small and limited in scope. To overcome this, we introduce *SemiSynCXR*, a framework designed to automatically generate semi-synthetic localization datasets. *SemiSynCXR* operates by inpainting specific radiological findings into real, healthy CXRs at anatomically plausible locations, which allows for the output of both the edited image and the ground-truth bounding box for each finding. *SemiSynCXR*-generated CXRs effectively augment existing localization datasets, yielding relative $mAP_{10:70}$ gains of up to 11% on in-domain and 21% on out-of-domain data, thereby mitigating data scarcity and improving generalization. Comprehensive quantitative and qualitative evaluations show that our framework achieves an overall AUROC of 0.78 and $mAP_{10:70}$ of 0.45, comparable to fully synthetic benchmarks. These results confirm that the generated findings are realistic and accurately localized, establishing *SemiSynCXR* as a practical solution for the generation of CXR finding localization datasets. Code available at the SemiSynCXR GitHub Repository.
**Keywords:** Semi-synthetic CXRs, Inpainting, Diffusion models

## 1. Introduction

A chest X-ray (CXR) is a fast, effective, and rather inexpensive aid for diagnosing and monitoring thoracic conditions (Mayo Clinic, 2024). Due to their prevalence in clinical practice, CXRs are a prominent image source for medical deep learning applications. Large datasets for radiological finding classification are common, often supported by automated

Figure 1: We propose *SemiSynCXR*, a framework that generates semi-synthetic CXRs by inpainting radiological findings into healthy images at plausible anatomical locations. *SemiSynCXR* automatically produces both the edited image and a precise bounding box, directly addressing data scarcity and class imbalance of existing localization datasets.

labeling, whereas curating datasets for localization and segmentation requires costly, time-intensive manual annotation by medical experts. As a result, these datasets are not only scarce, but also tend to be small and limited in scope.

To date, only eight publicly available CXR datasets provide bounding boxes for more than one radiological finding (Wang et al., 2017; Boecking et al., 2022b; Nguyen et al., 2022; Pham et al., 2023; Bigolin Lanfredi et al., 2022; Coelho de Castro et al., 2025; Fan et al., 2024; Liu et al., 2020), and two provide segmentation masks (Liu et al., 2020; Saporta et al., 2022). Of these eight localization datasets, only two contain over 10 000 scans[1]. Moreover, these datasets suffer from class imbalance. Even in the largest dataset, CXR-AL14 (with about 165 000 images), some common thoracic findings have low representation. For instance, atelectasis is found in only 0.2% of its images.

To overcome data scarcity in medical imaging, data synthesis offers a compelling solution (Voetman et al., 2023; Khosravi et al., 2024; Ktena et al., 2024). Driven by advances in high-quality image generation models like latent diffusion models (Rombach et al., 2022), existing studies have demonstrated the potential of generating synthetic CXRs (Bluethgen et al., 2025; Weber et al., 2023; Han et al., 2024; Huang et al., 2024). However, they largely focus on creating unlabeled images, image-text pairs, or images with only global classification labels, leaving the generation of much-needed localization datasets still unaddressed.

---

1. For comparison, PASCAL VOC (20 classes, 22 591 images) is considered one of the smallest widely used benchmarks for object detection in computer vision.

Moreover, obtaining the finding's exact positional information is a significant challenge when generating localization datasets using fully-synthetic image generation. Since the generative model implicitly determines the finding placement, its precise location often remains unknown. We propose semi-synthetic image generation as an alternative, as illustrated in Figure 1. By using automated image editing to inpaint findings into healthy CXRs, we can explicitly define the finding's location using conditioning masks. Thereby, the process inherently guarantees ground-truth bounding box annotations for every generated image, directly overcoming a core limitation of fully-synthetic approaches.

Our contributions are summarized as follows:

- We introduce *SemiSynCXR*, a framework for automatically generating semi-synthetic CXRs with radiological findings. The framework's core strength is its ability to provide the generated image with intrinsically matching, precise bounding boxes at scale.

- For this, we develop an automated mask generation method for *SemiSynCXR* that places findings at anatomically plausible locations based on real-world spatial distributions. *SemiSynCXR* further leverages existing diffusion models for inpainting findings into healthy CXR images, obviating the need for training new models.

- Using *SemiSynCXR*, we create a semi-synthetic dataset for CXR finding localization. Augmenting real training data with our generated samples significantly improves object detection performance and robustness, effectively mitigating data scarcity.

- Extensive quantitative and qualitative evaluations confirm that the generated findings are realistically placed and that the resulting CXRs resemble real images.

## 2. Related Work

### 2.1. Chest Radiological Findings Localization Datasets

The largest publicly available datasets for chest radiological findings are CXR-AL14 and VinDr-CXR. CXR-AL14 contains 165 988 posterior-anterior (PA) CXRs with bounding box annotations for 14 common abnormalities. These annotations were created through a "human-in-the-loop" process, where expert radiologists reviewed and corrected initial model annotations. VinDr-CXR is a smaller dataset of 18 000 PA CXRs, with manual local annotations for 22 critical findings. Despite their size, these datasets present significant limitations that could hinder model performance and robustness. Both originate from a very small number of institutions (one hospital in China for CXR-AL14, and two in Vietnam for VinDr-CXR), leading to low diversity in patient demographics, imaging equipment, and clinical workflows. This raises concerns regarding generalization to unseen clinical settings. Additionally, these datasets suffer from severe class imbalance, meaning some radiological findings are less frequently represented in the dataset, making their reliable detection challenging for models. More existing datasets are detailed in Appendix B.

### 2.2. Text- and Mask-Conditioned Diffusion Models for Editing

Using masks alongside text is an effective technique for guiding image editing, enabling precise, controlled changes without inadvertently altering adjacent areas (Huang et al.,

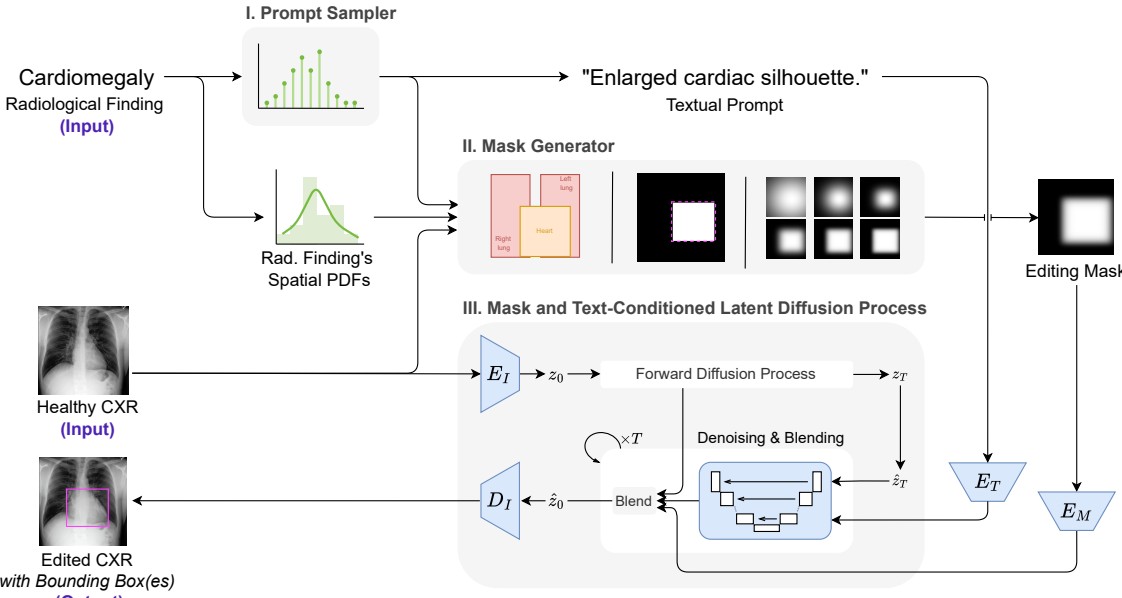

Figure 2: Overview of *SemiSynCXR*. Based on a real, healthy CXR and a specific finding, our method samples a textual prompt and generates a plausible spatial mask to guide the editing process. A latent diffusion model then uses the chest X-ray, prompt, and mask to inpaint the finding. The resulting semi-synthetic CXR and bounding boxes (derived from the conditioning mask) are used as targets for localization tasks.

2025). Hence, this approach is highly appealing for medical imaging applications, with several studies demonstrating its potential (Rouzrokh et al., 2023; Jin et al., 2024; Pérez-García et al., 2024; Hashmi et al., 2024; Chu et al., 2025). Some of these approaches require specific training or fine-tuning of the diffusion model, such as those by Rouzrokh et al. (2023) for brain tumor editing or Jin et al. (2024) for background alteration. Additionally, they often rely heavily on accurate, user-defined segmentation masks, which limits their scalability. Other approaches, conversely, eliminate the need for additional supervision by leveraging the iterative nature of the diffusion process itself. These frequently use multi-stage or multi-masking strategies, as seen in RadEdit (Pérez-García et al., 2024), XReal (Hashmi et al., 2024), and ChestX-rays_Mpe (Chu et al., 2025).

Reducing artifacts and preserving anatomical accuracy remain major challenges in CXR editing. Most existing methods either depend on user-defined masks, which inherently hinders large-scale applicability, or use broad anatomical regions, which limits the precision necessary to define ground-truth bounding boxes for specific findings. Additionally, comprehensive reporting on editing quality at the radiological finding level is often absent. Crucially, to the best of our knowledge, no prior studies have reported analyses on supplementing training data with (semi-)synthetic images specifically for chest radiological finding localization. This highlights an unaddressed gap that our study aims to fill.

## 3. Methodology

Our *SemiSynCXR* framework generates semi-synthetic CXRs by inpainting specific radiological findings into healthy images. The process, illustrated in Figure 2, begins with a real, healthy CXR and a target radiological finding. We then sample a textual prompt and generate a plausible spatial mask to guide the placement, drawing on real-world spatial distributions. Conditioned on these, a latent diffusion model (either RadEdit (Pérez-García et al., 2024) or RoentGen (Bluethgen et al., 2025)) then inpaints the finding into the healthy image. The output is a new semi-synthetic CXR and its ground-truth bounding boxes, directly derived from the conditioning mask. Seven findings are currently supported: Atelectasis, Cardiomegaly, Consolidation, Edema, Lung Opacity, Pleural Effusion, and Pneumothorax.

### 3.1. Datasets

*SemiSynCXR* itself does not require any training data; however, we leverage the following datasets to source healthy images, create textual prompts, and guide mask generation:

- MIMIC-CXR-JPG (Johnson et al., 2019b, 2024; Goldberger et al., 2000) contains 377 110 CXRs derived from MIMIC-CXR (Johnson et al., 2019a). It provides healthy CXRs (Section 3.2), and radiology reports to create textual prompts (Section 3.3).

- MS-CXR (Boecking et al., 2022b,a; Goldberger et al., 2000), constructed from MIMIC-CXR, consists of 1 162 image-sentence pairs with bounding boxes. We use it as source for medical texts to create textual prompts (Section 3.3), and for estimating the expected spatial distribution of different findings in the lung (Section 3.4).

- CheXmask (Gaggion et al., 2024b,a; Goldberger et al., 2000) comprises 657 566 anatomical segmentation masks, generated by a HybridGNet (Gaggion et al., 2021), for CXR datasets including MIMIC-CXR-JPG and VinDr-CXR. It provides the lung and heart segmentation masks for the mask generation and CXR editing (Sections 3.4 and 3.5).

- Chest ImaGenome (Wu et al., 2021; Goldberger et al., 2000), an anatomy-centered scene graph dataset from MIMIC-CXR, includes 29 chest anatomical locations and a manually annotated subset for 500 unique patients (gold standard dataset). We use this subset to source anatomical reference locations for mask generation (Section 3.4).

For evaluation, we incorporate the VinDr-CXR dataset (Nguyen et al., 2022, 2021; Goldberger et al., 2000). VinDr-CXR consists of 15 000 PA CXRs for training and 3 000 for testing (further split to obtain a validation set). Notably, this dataset captures patient demographics and imaging protocols distinct from those of MIMIC-CXR.

### 3.2. Sourcing Real, Healthy Chest X-rays

Instead of fully synthetic generation (Bluethgen et al., 2025; Lee et al., 2023; Han et al., 2024; Huang et al., 2024; Hashmi et al., 2024; Chu et al., 2025), we use a semi-synthetic approach: inpainting findings into real, healthy CXRs. Thus, *SemiSynCXR* preserves authentic image characteristics in unaffected regions while enabling precise control over finding placement, which inherently guarantees knowing the ground-truth bounding boxes.

Healthy chest X-rays are sampled from MIMIC-CXR-JPG using the following criteria: (i) the image must be a `posterior-anterior (PA)` view with `erect` patient posture; (ii) it must not be included in the MS-CXR dataset; (iii) it must be labeled as `No Finding` and negative for `Support Devices`, according to the CheXpert (Irvin et al., 2019) annotations; and (iv) it must be classified as negative for all relevant radiological findings by the XVR DenseNet-121 model (Cohen et al., 2022). Based on these criteria, we identified 24 555 eligible CXRs for sampling.

### 3.3. Sampling Textual Prompts

To steer the diffusion model toward the desired radiological finding, we condition the model on a textual prompt. For each finding, we curate a set of phrases derived from medical texts (Appendix C). We then sample a prompt from the corresponding set during editing.

Phrases are sourced from MIMIC-CXR radiology reports (exclusively associated with a single CheXpert finding) and MS-CXR textual descriptions (also associated with a single finding). From MIMIC-CXR radiology reports, we extract the *Findings*, *Impression*, or last section following Johnson et al. (2022, 2018). Only phrases observed more than once are retained and simplified using the `gpt-oss-20b` language model (OpenAI et al., 2025), removing mentions of size and severity, which are instead controlled by the editing mask. After manual review of simplified phrases, we adjust the sampling probabilities to ensure balanced sampling: phrases from MIMIC-CXR and MS-CXR are equally likely to be selected.

### 3.4. Generating the Editing Mask

To outline the location for inpainting, we sample a mask conditioned on the target finding, the sampled prompt (Section 3.3), and the anatomical structures of the sampled healthy CXR (Section 3.2). More precisely, for lung-associated findings, we model the spatial distribution in relation to the lungs, while for cardiomegaly, we consider the cardiothoracic ratio (CTR–i.e., the ratio between the heart's and thorax's widths). Modeling the spatial distributions relative to these anatomical structures, rather than in pixel space, eliminates the need for image registration and exploits the fact that the heart and lungs are easily identifiable in CXRs (Gómez et al., 2020). Within *SemiSynCXR*, the lung and heart structures are identified using CheXmask-derived bounding boxes.

**Spatial Distribution Estimation** We estimate the (relative) center and size distributions of the findings' bounding boxes based on data provided in the MS-CXR dataset.

For each lung-related finding, we model the center and size either as multivariate (2D) log-normal distributions or as two independent univariate (1D) distributions, based on a Spearman's correlation test for independence. The 1D distributions are selected according to the residual sum of squares (RSS) goodness-of-fit criterion[2]. We additionally assume lateral symmetry for the left and right lungs and thus mirror the source data in the contralateral lung before computing the distributions. Although assumed to simplify the modeling process, note that lateral symmetry does not fully reflect reality: A two-sample

---

2. 1D considered distributions: normal, generalized extreme value, exponential, gamma, Pareto, log-gamma, lognormal, beta, Student's $t$, and uniform distribution.

Kolmogorov–Smirnov test revealed enough statistical evidence against symmetry for two (edema and pleural effusion) of the six considered lung-related findings.

For cardiomegaly, we estimate the distribution of the CTR. Cardiomegaly is conventionally defined as present if the CTR is greater than 0.5 on a PA CXR. Hence, anterior-posterior (AP) images are excluded during estimation as hearts appear enlarged in this projection.

**Mask Sampling** To generate the mask, we sample the relative center and size from the estimated distributions of the given finding and convert these into image coordinates using the lung/heart masks.

During sampling, we must consider several constraints to assure findings are plausibly placed. First, we constrain the sampled sizes so the final masks remain within the lung boundaries. Additionally, textual prompts may contain anatomy-specific references, such as "bibasilar atelectasis", which indicates the finding is located at the base of both lungs. To ensure alignment with the prompt, we estimate the bounding boxes of the lung's anatomical substructures (like the lung bases) relative to the full lung itself using the Chest ImaGenome's gold standard dataset. We then use these bounding boxes so the center of the sampled mask lies on the lung area indicated by the prompt. All constraints are enforced via probability distribution truncation, using inverse transform sampling for the 1D distributions and an efficient sampling method for truncated multivariate normal distributions (Brunzema and Kim, 2021; Botev, 2017) for the 2D distributions.

For *pleural effusion* and *cardiomegaly*, we developed specialized mask sampling methods. For pleural effusion, we use the full width of the lung (left or right) and sample the center $y$-coordinate and height of the mask, guaranteeing full coverage of the bottom of the lung's bounding box. For cardiomegaly, we sample from the CTR distribution, and compute the mask size from the sampled ratio using the lung and heart masks of the current X-ray. The mask's center is set as that of the heart's bounding box.

Finally, the masks are blurred using generalized Gaussian filters. Blurring is found to help reduce artifacts as inpainted findings blend more naturally into healthy CXRs.

### 3.5. Editing: Inpainting the Radiological Finding

We generate the final semi-synthesized CXR by editing the healthy CXR using a latent diffusion model conditioned on the sampled healthy image, sampled textual prompt, and generated editing mask. We employed either RoentGen (Bluethgen et al., 2025) or RadEdit (Pérez-García et al., 2024) as the latent diffusion model, depending on the radiological finding. Note that these models are used pre-trained, requiring no fine-tuning or training on bounding box data. While both models natively support conditioning on textual prompts, we extend RoentGen to additionally support mask conditioning.

In both models, we use the blending method (Avrahami et al., 2023) for mask conditioning, which leverages the iterative nature of the reverse diffusion process. We consider three different variations of blending: (i) blending between the latents from the forward and reverse diffusion process before denoising (blending before; used with RoentGen only), (ii) blending between the latents from the forward and reverse diffusion process after denoising (blending after; used with RoentGen only), and restricting the classifier-free guidance (CFG) to the editing area (CFG masking; used with both RoentGen and RadEdit).

### 3.6. Finding the Optimal Configuration

We explored different design configurations of our framework to identify the best setting for each of the seven radiological findings under analysis. Specifically, we varied the diffusion model (RoentGen or RadEdit), mask blurring parameters, number of steps with mask conditioning, and hyperparameters of the diffusion inference process (Appendix E).

For each design configuration, we generated 35 semi-synthetic chest X-rays. We then used these samples to select the optimal *SemiSynCXR* configuration per finding based on four metrics: (i) Area Under the ROC Curve (AUROC), using a DenseNet-121 trained on XRV-all (Cohen et al., 2022); (ii) Fréchet Inception Distance (FID), obtained with InceptionV3 (layer 2048) (Szegedy et al., 2016); (iii) CLIP Score, derived from the XRayCLIP model (Chen et al., 2024); (iv) Average Precision ($AP_{10:70}$), using an ensemble of YOLOv4 models (Wang et al., 2021; Vinokurov, 2021) trained on VinDr-CXR. We aggregated these metrics into a single selection score by computing the arithmetic mean.

### 3.7. Implementation

*SemiSynCXR*'s editing component is built upon the Stable Diffusion Inpainting pipeline from the HuggingFace `Diffusers` library (von Platen et al., 2022). RoentGen weights were provided by the authors (version dated December 31, 2023), while RadEdit weights were sourced directly from the HuggingFace Hub. Experiments were conducted on an NVIDIA RTX A6000 GPU and NVIDIA A40 GPUs. Further details are provided in Appendix E.

## 4. Experiments and Results

Using our *SemiSynCXR* framework, we generate a semi-synthetic localization dataset to supplement real training data for radiological finding localization in CXRs. By training object detectors on this combined data, we first assess the impact of our generated CXRs as supplementary training data (Section 4.1). We then evaluate the quality of our generated images (Section 4.2).

### 4.1. SemiSynCXR for Supplementary Training Data Generation

To investigate how *SemiSynCXR*-generated images improve object detection training, we created a finding localization dataset of 35 000 samples and use it as an extension of the VinDr-CXR dataset (15 000 real images). Table 1 shows the overall results of training YOLO11n and YOLOv8n object detectors (Jocher and Qiu, 2024; Jocher et al., 2023) on VinDr-CXR alone or supplemented with subsets of our generated dataset. We report the $mAP_{10:70}$, averaged over IoU thresholds 0.1 to 0.7, and $mAP_{30}$ on the VinDr-CXR test set (in-distribution) and MS-CXR dataset (out-of-distribution). Statistical analysis of the $mAP_{10:70}$ differences was assessed using a Wilcoxon Signed-Rank test with the alternative hypothesis that the median of these differences is greater than zero ($H_1$ : median($mAP_{\text{Augmented}} - mAP_{\text{Baseline}}$) > 0). Pairwise samples were generated using fully random and stratified bootstrap resampling ($N = 100$, $N = 1\,000$).

Supplementing with our data improves the overall performance by up to 11% in $mAP_{10:70}$ on VinDr-CXR and up to 21% in $mAP_{10:70}$ on MS-CXR compared to training solely on real samples. For VinDr-CXR, using all 35k generated samples (1:2.3 real-to-synthetic ratio)

Table 1: Effect of *SemiSynCXR*-generated CXRs as supplementary training data for finding localization. YOLO11n and YOLOv8n detectors are trained on VinDr-CXR supplemented with varying quantities of our semi-synthetic images. We report $mAP_{10:70}$ (IoU 0.1-0.7) and $mAP_{30}$ on VinDr-CXR test set (in-distribution) and MS-CXR (out-of-distribution). Augmenting with our data increases $mAP_{10:70}$ by up to 11% (VinDr-CXR) and 21% (MS-CXR), confirming that *SemiSynCXR* helps to effectively address data scarcity and improve model generalization.

| Model | Training Data | | VinDr-CXR | | MS-CXR | |
|---|---|---|---|---|---|---|
| | Real | Synth | $[mAP_{10:70}$ (%) ↑] | $[mAP_{30}$ (%) ↑] | $[mAP_{10:70}$ (%) ↑] | $[mAP_{30}$ (%) ↑] |
| YOLO11n | 15k | – | 21.9 | 26.5 | 9.5 | 13.5 |
| | 15k | 7k | 22.9 (+ 5%) | 28.7 (+ 8%) | 10.3 (+ 8%) | 14.5 (+ 7%) |
| | 15k | 17.5k | 22.5 (+ 3%) | 27.5 (+ 4%) | **10.8** (+13%) | **15.2** (+13%) |
| | 15k | 35k | **24.2** (+11%) | **29.8** (+12%) | 9.6 (+ 0%) | 13.8 (+ 2%) |
| YOLOv8n | 15k | – | 21.9 | 26.4 | 9.4 | 13.2 |
| | 15k | 7k | 22.9 (+ 5%) | 27.8 (+ 5%) | 9.6 (+ 3%) | 14.1 (+ 7%) |
| | 15k | 17.5k | 23.6 (+ 8%) | 28.9 (+ 9%) | **11.3** (+21%) | **15.6** (+18%) |
| | 15k | 35k | **23.7** (+ 8%) | **29.1** (+10%) | 10.7 (+14%) | 14.6 (+11%) |

Table 2: Per-finding effect of SemiSynCXR-generated CXRs as supplementary training data on out-of-distribution localization performance. We report $AP_{10:70}$ on MS-CXR, using YOLO11n and YOLOv8n detectors trained on VinDr-CXR with varying quantities of our semi-synthetic images. Augmenting with our data improves localization performance across nearly all findings, most notably for Edema (up to $3\times$ gain) and Pneumothorax (up to 97% relative increase).

| Model | Training Data | | MS-CXR Localization [$AP_{10:70}$ (%) ↑] | | | | | | | |
|---|---|---|---|---|---|---|---|---|---|---|
| | Real | Synth | Atel. | Cmgl. | Cnls. | Edema | Opac. | P.Eff. | Pneum. | Avg. |
| YOLO11n | 15k | – | **1.9** | **42.7** | 11.4 | 1.3 | 1.6 | 4.4 | 3.5 | 9.5 |
| | 15k | 7k | 1.4 | 42.6 | 12.4 | 1.9 | 2.1 | **4.7** | 6.7 | 10.3 |
| | 15k | 17.5k | 1.6 | **42.7** | **12.6** | **5.2** | 1.8 | 4.5 | **6.9** | **10.8** |
| | 15k | 35k | 1.8 | 37.6 | 10.3 | 4.6 | **2.3** | 4.6 | 6.0 | 9.6 |
| YOLOv8n | 15k | – | **2.5** | 42.3 | 8.5 | 1.4 | 2.0 | 3.6 | 5.4 | 9.4 |
| | 15k | 7k | 1.7 | 39.7 | 11.3 | 2.6 | 1.8 | 4.2 | 6.1 | 9.6 |
| | 15k | 17.5k | 1.7 | **44.9** | **13.9** | 4.0 | **2.3** | **5.3** | **6.8** | **11.3** |
| | 15k | 35k | 1.6 | 42.5 | 12.8 | **5.0** | 2.0 | 4.8 | 6.4 | 10.7 |

leads to the best performance. For MS-CXR, however, the peak performance is achieved when supplementing with 17.5k samples (1:1.7 ratio), suggesting that adding an excessive number of semi-synthetic samples might sometimes introduce bias. This scaling behavior indicates that the optimal real-to-synthetic ratio is likely dataset-dependent; however, a comprehensive ablation study to determine these precise thresholds remains a subject for future investigation.

Table 3: Factual correctness of *SemiSynCXR*-generated CXRs. We benchmark detectability (AUROC using a DenseNet121) and localization accuracy ($AP_{10:70}$ using an ensemble of YOLOv4s trained on VinDr-CXR) against real and fully synthesized CXRs. Our approach yields findings detectable at levels comparable to, or superior to, fully synthesized CXRs. High AUROC scores for cardiomegaly and pleural effusion suggest these semi-synthetic CXRs closely resemble prototypical clinical cases. Strong $AP_{10:70}$ scores confirm *SemiSynCXR* successfully also produces realistic, well-localized findings.

| Model | Radiological Finding | | | | | | | |
|---|---|---|---|---|---|---|---|---|
| | Atel. | Cmgl. | Cnls. | Edema | Opac. | P. Eff. | Pneum. | Avg. |
| **Classification** [AUROC ↑] | | | | | | | | |
| XVR's benchmark (Cohen et al., 2022) (real) | 0.88 | 0.88 | 0.91 | 0.92 | 0.86 | 0.92 | 0.81 | 0.88 |
| RoentGen (Bluethgen et al., 2025) (synthetic) | 0.76 | 0.82 | 0.69 | 0.85 | 0.74 | 0.90 | 0.61 | 0.76 |
| CXRL (Han et al., 2024) (synthetic) | 0.86 | 0.88 | 0.94 | 0.89 | 0.70 | 0.77 | 0.88 | 0.81 |
| LLM-CXR (Lee et al., 2023) (synthetic) | 0.81 | 0.78 | 0.82 | 0.81 | 0.83 | 0.82 | 0.75 | 0.80 |
| Chest-Diffusion (Huang et al., 2024) (synthetic) | 0.70 | 0.73 | 0.63 | 0.79 | 0.65 | 0.85 | 0.57 | 0.70 |
| SemiSynCXR (ours) | 0.72 | 0.98 | 0.73 | 0.82 | 0.67 | 0.97 | 0.58 | 0.78 |
| **Localization** [$AP_{10:70}$ (%) ↑] | | | | | | | | |
| VinDr-CXR (Nguyen et al., 2022) (real) | 6.97 | 76.04 | 19.32 | – | 5.05 | 63.47 | 29.68 | 30.57 |
| SemiSynCXR (ours) | 14.64 | 97.26 | 52.56 | – | 36.58 | 58.34 | 8.80 | 44.70 |

At the finding level, augmenting with SemiSynCXR's samples improves localization performance across nearly all radiological findings (Table 2; Table 4, Appendix A.1). Specifically, findings that are difficult to automatically localize (e.g., pneumothorax) benefited most, whereas those with already high baseline accuracy (e.g., cardiomegaly) saw more modest gains. Atelectasis remains a challenge across both datasets, exhibiting low APs@10:70 and minimal response to augmentation, suggesting that its radiological features may not yet be fully captured by our current framework. The impact of augmentation varies both by radiological findings and the real-to-synthetic ratio employed.

Notably, the null hypothesis ($H_0$ : median $\leq 0$) was rejected at a confidence level of $\alpha = 0.05$ across all scenarios (in-distribution and out-of-distribution testing), confirming that the observed gains in $mAP_{10:70}$ are statistically significant. Overall, the results confirm that our framework serves as an effective solution to data scarcity while enhancing the generalization capability of object detection models.

## 4.2. Generation Quality

We quantitatively assess the factual correctness of our generated images using classification models, following common practice, and localization models. Finding detection (whether desired finding is successfully inpainted into healthy image) is measure by the AUROC from a DenseNet-121 classifier trained on XRV-all (Cohen et al., 2022). Accurate placement (localization) is measured using the $AP_{10:70}$ from an ensemble of YOLOv4 models (Wang et al., 2021; Vinokurov, 2021) trained on VinDr-CXR (Nguyen et al., 2022), averaging over IoU thresholds 0.1 to 0.7. Edema is excluded from the $AP_{10:70}$ metric due to its underrepresenta-

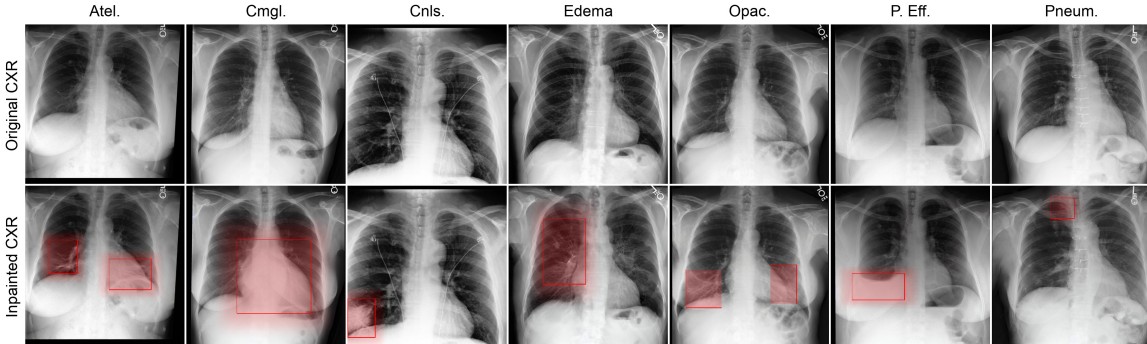

Figure 3: Examples of *SemiSynCXR*-generated CXRs. We show the real, healthy chest X-rays (top) and edited versions (bottom). The red outlines correspond to the conditioning masks alongside their non-blurred version, which serve as training targets for localization models. Additional examples can be found in Appendix A.5.

tion in VinDr-CXR, and bounding boxes for cardiomegaly and pleural effusion are rescaled to account for distributions shifts between our editing masks and VinDrCXR's annotations. We benchmark model performance on *SemiSynCXR* samples against their performance on real CXRs and on fully synthesized CXRs.

Results, presented in Table 3, show that our approach produces findings that are detectable by the classifier at levels comparable to, or even better than, fully synthesized CXRs, and strong $AP_{10:70}$ scores confirm successful localization. At the individual findings level, cardiomegaly and pleural effusion achieve particularly strong performance, suggesting that these findings resemble prototypical clinical cases. We attribute this success to their consistent anatomical placement (at the heart and lung bases, respectively), claim that is further supported by per-finding localization results presented in Section 4.1. By contrast, pneumothorax shows comparatively weaker performance, likely because this finding often affects large portions of the lung beyond the localized inpainting region, making realistic generation challenging when constrained to narrow masks. Overall, the generated samples demonstrate high factual correctness for most findings, confirming their suitability as training data; however, performance differences compared to real CXRs indicate room for further improvement.

While high AUROC scores are necessary, they are not sufficient for guaranteeing image realism (e.g., they can result from a model over-exaggerating certain pathological features). This is why our quality evaluation also includes visual alignment with MIMIC-CXR using Fréchet Inception Distance (FID) and visual-text alignment with conditioning prompts using CLIPScore (Appendix A.2). Our approach achieves comparable performance to most state-of-the-art methods while uniquely providing ground-truth bounding boxes. Finally, a qualitative study by three medical experts on 140 randomly selected CXRs (70 generated from a pool of 35k images, 70 real from a pool of around 10k images) found that, on average, 36% of generated images were judged as real (compared to 71% of real images judged as real), and the intended finding was correctly recognized in 54% of generated cases (vs. 28% in real images) (Appendix A.3). Examples of the generated CXRs are shown in Figure 3.

## 5. Discussion and Conclusions

We introduce *SemiSynCXR*, a framework for automatically generating localization datasets for chest radiological findings. Our framework's core strength lies in its ability to provide the generated images with intrinsically matching, precise bounding boxes at scale. Extensive evaluations confirm that the quality and realism of the edited images are comparable to fully synthetic data, while simultaneously demonstrating their utility as a training data augmentation source. Our findings suggest that *SemiSynCXR* provides a practical and effective solution to addressing data scarcity in medical imaging.

Nonetheless, certain limitations remain. Editing quality is constrained by both the capabilities of the underlying diffusion models and the effectiveness of our mask generation strategy. Although model performance could be enhanced through fine-tuning, improving mask conditioning is more challenging. Specifically, the bounding-box-constrained editing approach, while beneficial for maintaining structural integrity and providing precise ground truth annotations, may fail to fully capture the changes when findings extend beyond their localized bounding boxes, potentially struggling to represent diffuse conditions accurately. Promising future directions for addressing these limitations include the implementation of iterative mask relaxation and the use of anatomical region bounding boxes. Additionally, our current framework does not explicitly account for finding size and severity, which are instead influenced by the sampled editing mask. Integrating explicit control over these attributes would enable more nuanced generation guidance.

A larger-scale medical expert study is also essential to further validate the clinical realism and utility of the generated data. Similarly, large-scale multi-center evaluations would be beneficial for identifying potential dataset biases. Among the framework's components (i.e., healthy input CXRs, editing-conditioning elements, and the underlying model), the healthy input X-rays possess the highest potential as a source of dataset-specific biases. Consequently, the inclusion of more diverse CXR datasets within these components is essential to strengthen the framework's robustness across varied clinical settings.

Expanding SemiSynCXR to support additional radiological findings, such as calcifications, fractures, and nodules, is a natural extension that primarily requires the probability distributions of the bounding boxes for such findings. Beyond chest X-rays, this approach is applicable to other clinical conditions characterized by rather focal, non-diffuse manifestations. Examples include tumors and lesions in oncology, aneurysms and hemorrhages in cardiovascular imaging, and drusen in ophthalmology. The framework is also potentially adaptable to other 2D and 3D imaging modalities, as well as to time series of images. Such extensions require an existing pretrained text-conditioned diffusion model (or data to train one) and the probability distributions of the bounding boxes.

A strategic advantage of our framework is its ability to leverage pretrained models, thereby benefiting from ongoing advancements in the field without the need for training from scratch. Even if the underlying model must be trained, our approach remains viable since training relies only on image-text pairs (derivable from vision-language models) rather than manual bounding box annotations. All of these factors demonstrate our framework's broad applicability and potential for continued development. Finally, we see great opportunities in modeling rare clinical cases, which could significantly enhance the robustness of automated object detectors and provide substantial clinical utility.

## Acknowledgments

Andrea Posada and Martin J. Menten are funded by the German Research Foundation under project 532139938.

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

# Appendix A. Supplemental Results

## A.1. SemiSynCXR for Supplementary Training Data Generation – Per-Finding Results

Table 4: Per-finding effect of SemiSynCXR-generated CXRs as supplementary training data on in-distribution localization performance. We report $AP_{10:70}$ on VinDr-CXR, using YOLO11n and YOLOv8n detectors trained on VinDr-CXR with varying quantities of our semi-synthetic images. Augmenting with our data improves localization performance across nearly all findings, with gains especially observed for Pneumothorax (up to 96% relative increase) and Consolidation (up to 35% relative increase).

| Model | Training Data | | VinDr-CXR Localization [$AP_{10:70}$ (%) ↑] | | | | | | |
|---|---|---|---|---|---|---|---|---|---|
| | Real | Synth | Atel. | Cmgl. | Cnls. | Opac. | P.Eff. | Pneum. | Avg. |
| YOLO11n | 15k | – | 1.9 | 64.1 | 14.4 | 2.4 | 31.7 | 16.8 | 21.9 |
| | 15k | 7k | **2.9** | 63.9 | 15.9 | 2.3 | **31.9** | 20.8 | 22.9 |
| | 15k | 17.5k | 2.8 | **66.5** | 16.3 | **2.7** | 30.2 | 16.5 | 22.5 |
| | 15k | 35k | 1.4 | 66.0 | **19.4** | 2.1 | 31.4 | **25.0** | **24.2** |
| YOLOv8n | 15k | – | **3.2** | 64.2 | 15.9 | 2.8 | 32.4 | 12.8 | 21.9 |
| | 15k | 7k | 1.6 | 64.4 | **16.2** | **3.4** | 30.8 | 21.4 | 22.9 |
| | 15k | 17.5k | 3.0 | **66.6** | 15.8 | 3.0 | **32.6** | 20.9 | 23.6 |
| | 15k | 35k | 2.2 | 64.8 | 15.4 | **3.4** | 31.3 | **25.1** | **23.7** |

## A.2. Visual and Visual-Text Alignment

We measure the visual alignment of our generated images with the MIMIC-CXR dataset using the Fréchet Inception Distance (FID) score, obtained with InceptionV3 (layer 2048) (Szegedy et al., 2016). To also assess the visual–text alignment, we employ the CLIPScore based on the CXR-CLIP model (You et al., 2023) and the XRayCLIP model (Chen et al., 2024)[3], which measures the similarity between each generated semi-synthetic image and its corresponding textual prompt.

The general and finding-level results are presented in Tables 5 and 6, respectively. Our approach achieves performance comparable to most existing methods, with only LLM-CXR yielding substantially better scores. However, a direct comparison of these scores across studies should be interpreted with caution due to: (i) the scores are computed on different subsets of finding classes, which alters the distribution of generated images, and (ii) the textual prompt distributions vary across studies, which can influence the CLIPScore. Overall, our method demonstrates image quality on par with state-of-the-art models, while uniquely providing ground-truth bounding boxes for the findings.

---

3. StanfordAIMIXrayCLIP__vit-l-14_laion2b-s32b-b82k

Table 5: Visual alignment (FID, InceptionV3) and visual–text (CLIPScore, CXR-CLIP) alignment of *SemiSynCXR*-generated CXRs. Our approach performs comparably to most existing methods, with only LLM-CXR achieving a notably better FID. These results, while broadly comparable, should be interpreted with caution due to differences in the findings considered and textual prompt distributions across studies. Overall, *SemiSynCXR* achieves competitive image quality while providing precise bounding boxes for radiological findings.

| Model | $FID_{InceptionV3}$ $\downarrow$ | $CLIPScore_{CXR-CLIP}$ $\uparrow$ |
|---|---|---|
| RoentGen (Bluethgen et al., 2025) (synthetic) | $64.60^{\ddagger}$ | $0.29^{\dagger\ddagger}$ |
| LLM-CXR (Lee et al., 2023) (synthetic) | $\mathbf{22.75}^{\ddagger}$ | $0.20^{\dagger\ddagger}$ |
| XReal (Hashmi et al., 2024) (synthetic) | $55.12^{\ddagger}$ | – |
| CXRL (Han et al., 2024) (synthetic) | – | $\mathbf{0.34}^{\ddagger}$ |
| SemiSynCXR (ours) | 63.99 | 0.30 |

$\dagger$: Scores as reported in CXRL (Han et al., 2024).
$\ddagger$: Limited comparability as these scores are averaged over a different set of finding classes.

Table 6: Visual alignment and visual–text alignment per radiological finding, measured using FID (InceptionV3) and CLIPScore (CXR-CLIP, XRayCLIP), respectively.

| | Atel. | Cmgl. | Cnls. | Edema | Opac. | P. Eff. | Pneum. | Avg. |
|---|---|---|---|---|---|---|---|---|
| **Visual Alignment** | | | | | | | | |
| $FID_{InceptionV3}$ $\downarrow$ | 61.62 | 73.22 | 63.24 | 67.09 | 58.57 | 61.34 | 62.84 | 63.99 |
| **Visual-Text Alignment** | | | | | | | | |
| $CLIPScore_{CXR-CLIP}$ $\uparrow$ | 0.28 | 0.28 | 0.34 | 0.14 | 0.29 | 0.42 | 0.36 | 0.30 |
| $CLIPScore_{XrayCLIP}$ $\uparrow$ | 0.21 | 0.23 | 0.21 | 0.22 | 0.21 | 0.27 | 0.27 | 0.23 |

Table 7: Qualitative evaluation of our generated CXRs. Three medical experts assessed 140 scans (70 generated, 70 real with findings). On average, 36% of generated images were judged as real, and the intended finding was correctly identified in 64% of generated cases, suggesting that many inpainted findings are recognizable.

| | Atel. | Cmgl. | Cnls. | Edema | Opac. | P. Eff. | Pneum. | Avg. |
|---|---|---|---|---|---|---|---|---|
| **Generated Images** | | | | | | | | |
| As Real [FPR $\uparrow$] | 0.10 | **0.50** | 0.30 | 0.40 | 0.37 | 0.47 | 0.40 | 0.36 |
| With Finding [TPR $\uparrow$] | 0.60 | 0.57 | **0.83** | 0.33 | 0.20 | 0.77 | 0.50 | 0.54 |
| **Real Images** | | | | | | | | |
| As Real [TPR $\uparrow$] | 0.77 | 0.63 | 0.80 | 0.77 | 0.63 | 0.70 | 0.70 | 0.71 |
| With Finding [TPR $\uparrow$] | 0.13 | 0.43 | 0.20 | 0.17 | 0.13 | 0.53 | 0.37 | 0.28 |

### A.3. Qualitative Assessment

We conducted a systematic study to qualitatively evaluate our generated CXRs. Three medical experts rated 140 randomly selected CXRs: 70 generated images (10 per finding) and 70 real images (10 per finding). For each CXR, raters identified (i) whether the image was real (realism) and (ii) which finding was present (finding recognition). Results, presented in Table 7, detail the False Positive Rate (FPR) for judging a generated image as real and the True Positive Rate (TPR) for correctly identifying the intended finding. For reference, we further report the TPRs in real images for realism and finding recognition.

On average, 36% of the 70 generated images are judged as real (vs. 71% on real images), and the intended finding is correctly identified in 54% of the generated cases (vs. 28% on real images). Pleural effusion and cardiomegaly demonstrate high rates for both realism and finding recognition. Atelectasis, consolidation, and pneumothorax also have high finding recognition rates, though there is room for improvement in their realism. Overall, these results suggest that many inpainted findings are recognizable, even so image generation artifacts may remain.

### A.4. Ablation Studies

We perform an ablation study comparing editing pipelines (diffusion model with blending strategy – Section 3.5) under different mask blurring parameters. Specifically, we evaluated four pipelines: RoentGen with blending before, RoentGen with blending after, RoentGen with CFG masking, and RadEdit with CFG masking. The overall generation quality score, as defined in Section 3.6, is presented in Figure 4. RoentGen generally achieves better or comparable performance to RadEdit, except for pneumothorax and consolidation, where RadEdit is notably superior. The choice of blending strategy has only a minor impact. Additional mask conditioning results are presented in Figures 5 and 6.

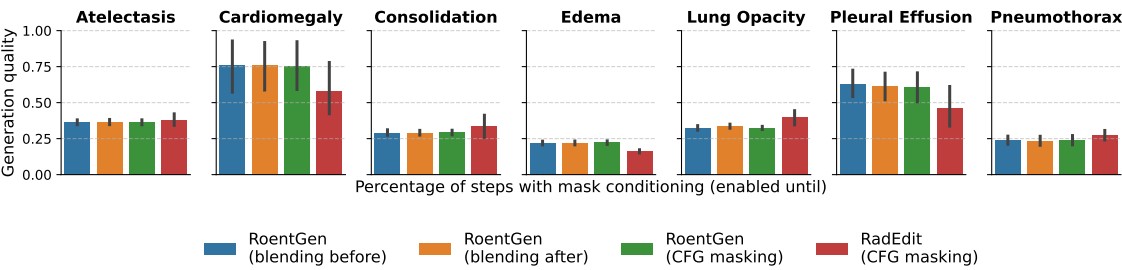

Figure 4: Ablation study on the different editing pipelines (RoentGen with blending before, RoentGen with blending after, RoentGen with CFG masking, and RadEdit with CFG masking). For each setting, we consider multiple mask blurring parameters shown as uncertainty intervals (Section 3.4) and compute the overall generation quality score following Section 3.6. We found that for most findings RoentGen generally performs better than or on par with RadEdit while the blending pipeline does not have a huge impact. However, note that for pneumothorax and consolidation RadEdit performs notably better.

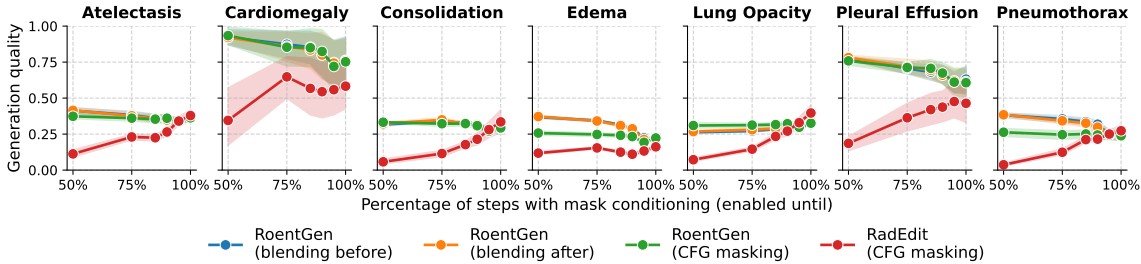

Figure 5: Ablation study on different blending pipelines and mask conditioning settings. We consider four different pipelines (RoentGen with blending before, RoentGen with blending after, RoentGen with CFG masking, and RadEdit with CFG masking) and vary the number of steps where mask conditioning is used. We enable mask conditioning for a specified percentage of steps ($x$-axis) before dropping it. For setting, we consider multiple mask blurring parameters (uncertainty intervals in the graph) and compute the overall generation quality score, following Sec. 3.6.

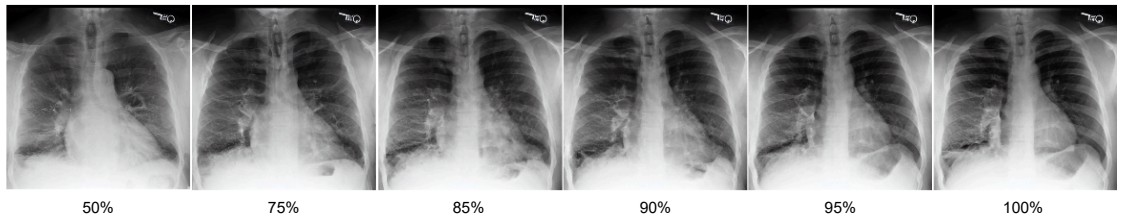

Figure 6: Example on the effect of using mask conditioning for a specified percentage of steps when inpainting edema with our approach.

## A.5. Examples of CXRs Generated by *SemiSynCXR*

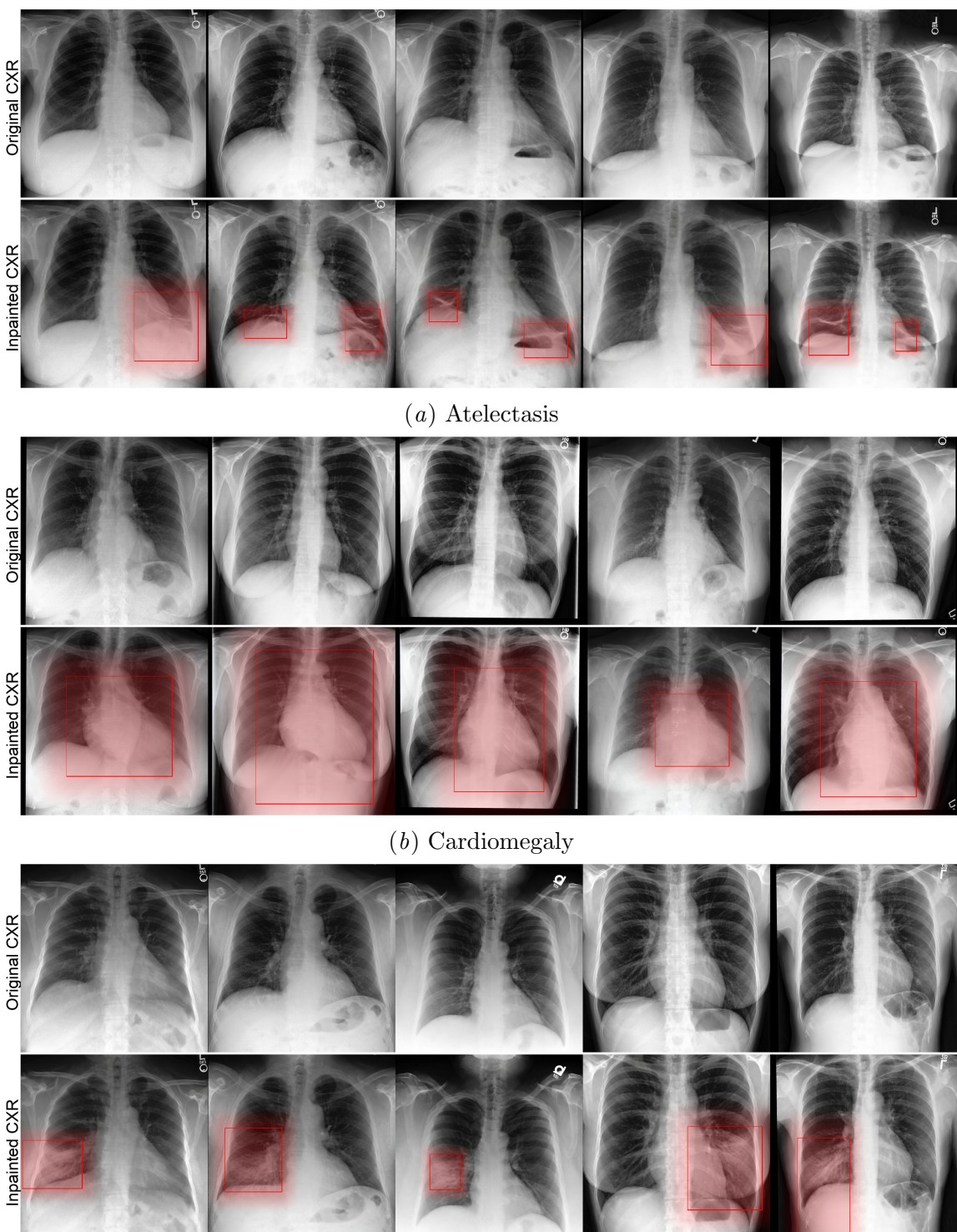

(*a*) Atelectasis

(*b*) Cardiomegaly

(*c*) Consolidation

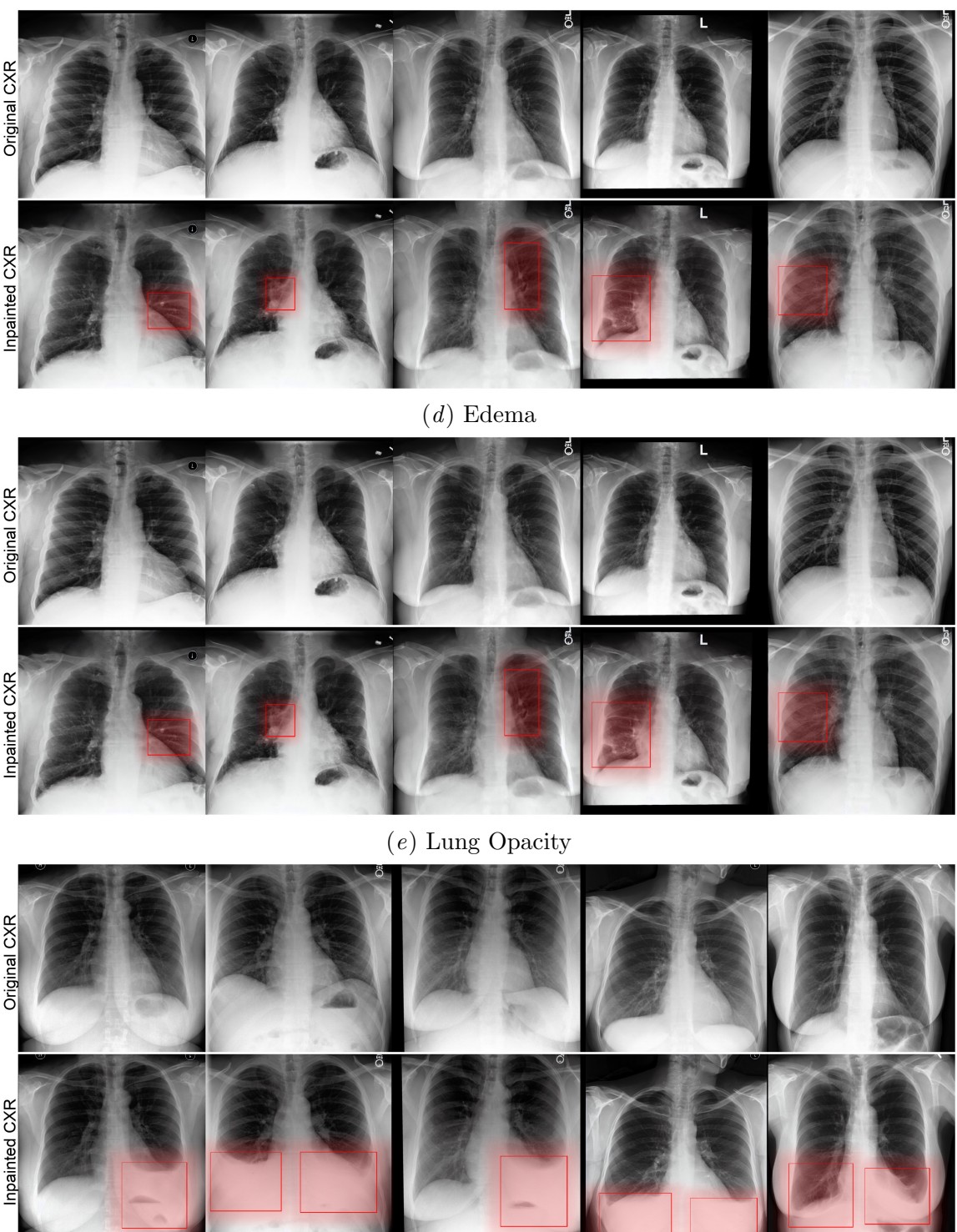

(*d*) Edema

(*e*) Lung Opacity

(*f*) Pleural Effusion

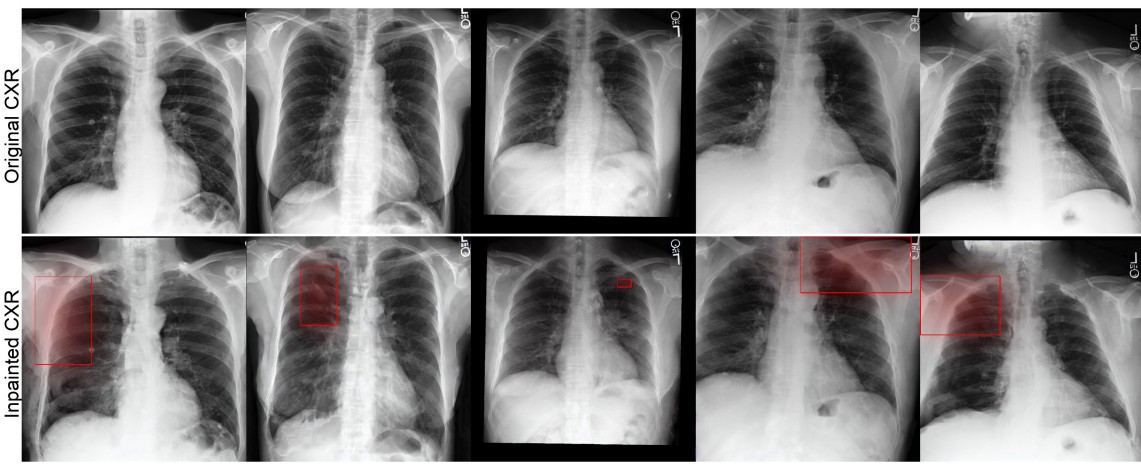

($g$) Pneumothorax

Figure 7: Additional examples of images generated by our *SemiSynCXR* framework, extending Figure 3. We show the real, healthy chest X-rays (top) and edited versions (bottom). The red outlines correspond to the conditioning masks alongside their non-blurred version, which serve as training targets (bounding boxes) for localization models.

## Appendix B. Existing Chest X-Rays Datasets for Radiological Finding Localization and Segmentation

We provide in Table 8 an overview of existing publicly available Chest X-ray (CXR) datasets for localization and segmentation of radiological findings. Datasets focusing on only one specific finding, such as those dedicated exclusively to pneumothorax or pulmonary nodules, have been excluded. Examples of the single-finding datasets omitted from this list include BIMCV-COVID19+, COVIDx CXR Dataset (COVID-Net) GRAZ+RSNA Pneumonia Detection Challenge, CANDID-PTX, PLCO Dataset (LIDC extension), SIIM-ACR Pneumothorax Segmentation.

Table 8: Overview of publicly available chest X-ray localization and segmentation datasets.

| Dataset | Targets | Size (No. CXRs) | Origin (Country/Dataset) | Available | Notes |
|---|---|---|---|---|---|
| NIH ChestX-ray14 | Disease labels (14), Disease bounding boxes (8; 880 with bounding boxes subset) | 112 120 frontal views; 880 with bounding boxes | United States | Kaggle, Google Cloud, and NIH download site | NLP-mined disease labels from radiology reports. |
| Vindr-CXR | Disease labels (6), Disease bounding boxes (22; train and 3k test) | > 100000; 18 000 with bounding boxes (15k train and 3k test) | Vietnam | PhysioNet | Training set labeled by three radiologists, test set labeled by consensus of 5 radiologists. |
| VinDr-PCXR | Disease labels (15), bounding boxes (36) | 9125 (7728 train and 1 397 test) | Vietnam | PhysioNet | Labeled by experienced radiologists. Pediatric dataset. |
| MS-CXR | Bounding boxes (8) with descriptions | 1162 | MIMIC-CXR | PhysioNet | Labels verified by board-certified radiologists. |
| PadChest-GR | Bounding boxes (24) with descriptions | 4555 frontal views | PadChest | BIMCV | Labeled by a team of 14 radiologists. Multilingual dataset (English and Spanish). |
| CXR-AL14 | Bounding boxes (14) | 165 988 | China | Dong Zhang[a] or cxr-al14.top | Human-in-the-loop labeling. |
| REFLACX + LATTE-CXR | Eye-tracking with dictated reports, ellipses (localization), anatomical bounding boxes | 3032 frontal views | MIMIC-CXR | PhysioNet | Eye-tracking and dictated reports from five radiologists. |
| ChestX-Det Dataset | Bounding boxes (13), segmentation masks (13) | 3578 | NIH ChestX-ray14 | GitHub | Labeled by board-certified radiologists. |
| CheXlocalize | Segmentation masks (10), keypoints | 902 | MIMIC-CXR | Standford AIMI | Labeled by board-certified radiologists. |

a. hszhangd@tmmu.edu.cn

## Appendix C. Textual Prompts

We present below the curated set of radiological finding-related phrases used as textual prompts for conditioning the diffusion model during the editing process.

| Rad. Finding | Textual Prompt | Probability ($p$) |
|---|---|---|
| Atelectasis | Bibasilar atelectasis. | 0.6406 |
| | Left basilar atelectasis. | 0.1647 |
| | Basilar atelectasis. | 0.0380 |
| | Bibasilar subsegmental atelectasis. | 0.0341 |
| | Right basilar atelectasis. | 0.0380 |
| | Left lower lobe atelectasis. | 0.0180 |
| | Atelectasis in the lung bases. | 0.0106 |
| | Left basilar subsegmental atelectasis. | 0.0053 |
| | Streaky bibasilar atelectasis. | 0.0042 |
| | Subsegmental atelectasis. | 0.0042 |
| | Linear bibasilar atelectasis. | 0.0063 |
| | Atelectasis. | 0.0158 |
| | Left lower lobe collapse. | 0.0032 |
| | Right lower lobe atelectasis. | 0.0021 |
| | Right basilar subsegmental atelectasis. | 0.0042 |
| | Patchy bibasilar atelectasis. | 0.0063 |
| | Right upper lobe collapse. | 0.0022 |
| | Right middle lobe collapse. | 0.0022 |
| Cardiomegaly | Cardiomegaly. | 0.7846 |
| | Enlarged cardiac silhouette. | 0.1940 |
| | Enlargement of the cardiac silhouette. | 0.0154 |
| | Prominent cardiac silhouette. | 0.0018 |
| | Enlarged heart. | 0.0042 |
| Consolidation | Left lower lobe consolidation. | 0.3064 |
| | Right lower lobe consolidation. | 0.2401 |
| | Patchy consolidation in the mid left lung. | 0.0704 |
| | Patchy consolidation in the right lung. | 0.0704 |
| | Patchy consolidation in the right lower lobe. | 0.1232 |
| | Left consolidation. | 0.0352 |
| | Patchy bilateral pulmonary consolidations. | 0.0352 |
| | Bilateral consolidations. | 0.0340 |
| | Right middle lobe consolidation. | 0.0511 |
| | Right upper lobe consolidation. | 0.0340 |
| Edema | Pulmonary edema. | 0.7310 |
| | Interstitial pulmonary edema. | 0.1333 |
| | Interstitial edema. | 0.1023 |
| | Edema. | 0.0175 |
| | Peribronchial cuffing consistent with pulmonary edema. | 0.0159 |
| Lung Opacity | Right lower lobe infiltrate. | 0.1635 |
| | Right lower lobe opacity. | 0.1499 |
| | Left lower lobe opacity. | 0.1908 |
| | Bilateral lower lobe infiltrates. | 0.0681 |
| | Left lower lobe infiltrate. | 0.0681 |

| | | |
|---|---|---|
| | Patchy bilateral pulmonary opacities. | 0.0514 |
| | Patchy left lower lobe opacity. | 0.0681 |
| | Bibasilar opacities. | 0.0409 |
| | Patchy ground-glass opacities at the right lung base. | 0.0386 |
| | Left basilar opacity. | 0.0273 |
| | Right basilar opacity. | 0.0273 |
| | Lower lung opacity. | 0.0273 |
| | Patchy ground-glass opacities in the left lower lung. | 0.0257 |
| | Patchy bibasilar opacities. | 0.0530 |
| Pleural Effusion | Bilateral pleural effusions. | 0.3979 |
| | Right pleural effusion. | 0.2429 |
| | Left pleural effusion. | 0.2351 |
| | Right effusion. | 0.0388 |
| | Bilateral effusions. | 0.0310 |
| | Right-sided pleural effusion. | 0.0284 |
| | Left-sided pleural effusion. | 0.0207 |
| | Left effusion. | 0.0052 |
| Pneumothorax | Right apical pneumothorax. | 0.3472 |
| | Left apical pneumothorax. | 0.3208 |
| | Right pneumothorax. | 0.1774 |
| | Left pneumothorax. | 0.1245 |
| | Pneumothorax. | 0.0151 |
| | Apical pneumothorax. | 0.0075 |
| | Bilateral pneumothoraces. | 0.0075 |

# Appendix D. Radiological Findings' Spatial Distributions

Table 10: Probability distributions for the spatial modeling of radiological finding bounding boxes. Parameters for the 1D best-fit distributions correspond to the location-scale form of the distribution in regard.

| Radiological Finding | Bounding Box Element | Distribution | Parameters | Score |
|---|---|---|---|---|
| Atelectasis | Center (x-axis) | Beta | $a : 194.8522$, $b : 78.1808 \times 10^6$, loc $: -119.7766$, scale $: 66.6710 \times 10^6$ | $2.36 \times 10^{-4}$ |
| | Center (y-axis) | Log-gamma | $c : 0.7680$, loc $: 87.8522$, scale $: 6.9387$ | $1.07 \times 10^{-3}$ |
| | Width & height | Multivariate log-normal | $\boldsymbol{\mu} : [4.3618, 3.4926]$, $\boldsymbol{\Sigma} : [[0.0842, 0.0632], [0.0632, 0.2054]]$ | $-$ |
| Cardiomegaly | CTR | Gamma | $a : 40.4439$, loc $: 33.4765$, scale $: 0.6308$ | $4.66 \times 10^{-3}$ |
| Consolidation | Center (x-axis) | Log-normal | $s : 0.1733$, loc $: -24.9657$, scale $: 69.8613$ | $6.94 \times 10^{-5}$ |
| | Center (y-axis) | Beta | $a : 9.3284$, $b : 3.6820$, loc $: -32.9031$, scale $: 132.7595$ | $1.20 \times 10^{-4}$ |
| | Width & height | Multivariate log-normal | $\boldsymbol{\mu} : [4.1543, 3.6383]$, $\boldsymbol{\Sigma} : [[0.1449, 0.1393], [0.1393, 0.3113]]$ | $-$ |
| Edema | Center | Multivariate log-normal | $\boldsymbol{\mu} : [3.8485, 3.9856]$, $\boldsymbol{\Sigma} : [[0.0968, -0.0336], [-0.0336, 0.0529]]$ | $-$ |
| | Width & height | Multivariate log-normal | $\boldsymbol{\mu} : [4.2697, 3.9856]$, $\boldsymbol{\Sigma} : [[0.1678, 0.1776], [0.1776, 0.2681]]$ | $-$ |
| Lung Opacity | Center (x-axis) | Beta | $a : 8.3376$, $b : 4\,440.2773$, loc $: 4.3958$, scale $: 2\,2401.2265$ | $2.24 \times 10^{-4}$ |
| | Center (y-axis) | Log-gamma | $c : 0.7223$, loc $: 77.5134$, scale $: 10.4228$ | $1.89 \times 10^{-4}$ |
| | | log-normal | $\boldsymbol{\Sigma} : [[0.0968, -0.0336], [-0.0336, 0.0529]]$ | |
| | Width & height | Multivariate log-normal | $\boldsymbol{\mu} : [4.0257, 3.5337]$, $\boldsymbol{\Sigma} : [[0.1754, 0.1410], [0.1410, 0.3298]]$ | $-$ |
| Pleural Effusion | Center (y-axis) | GEV | $c : 0.6393$, loc $: 88.9743$, scale $: 7.3628$ | $6.34 \times 10^{-4}$ |
| | Height | Beta | $a : 1.2177$, $b : 3.5508$, loc $: 13.3016$, scale $: 22\,401.2265$ | $2.43 \times 10^{-5}$ |
| Pneumothorax | Center | Multivariate log-normal | $\boldsymbol{\mu} : [3.9222, 2.7920]$, $\boldsymbol{\Sigma} : [[0.277, -0.3239], [-0.3239, 1.0157]]$ | $-$ |
| | Width & height | Multivariate log-normal | $\boldsymbol{\mu} : [4.1561, 3.2241]$, $\boldsymbol{\Sigma} : [[0.1881, 0.0425], [0.0425, 0.4092]]$ | $-$ |

GEV: Generalized extreme value distribution
CTR: Cardiothoracic ratio.

## Appendix E. Experimental Setup

As part of our experiments, we explored several design variations for generating semi-synthetic chest X-rays. These variations contemplated the following:

- Text prompts: Class labels directly (e.g., "Atelectasis") or sampled class-related phrases (see Section 3.3 and Appendix C).

- Mask blurring: No blur and six blurring alternatives using generalized Gaussian filters with standard deviation based on the size of the mask and a factor $r$, i.e.,

$$\boldsymbol{\sigma} = (\lfloor 0.5 \times \text{size(bounding box)} \rfloor)/r.$$

  Specifically, we tested with $\beta = 2.0$ and $r \in \{0.5, 1.0, 2.0\}$, and $r = 0.1$ and $\beta \in \{4.0, 6.0, 8.0\}$.

- Mask conditioning extend: 85%, 90%, 95%, and 100% of the total number of steps of the reverse diffusion process.

- Hyperparameters of the diffusion process:

  - Number of steps: 1.0, 1.33, and 2.0 times the default number of parameters.
  - Classifier-free guidance scale: RadEdit with default, 12.0, 15.0, and 19.0; RoentGen with default, 6.0, 7.5, and 10.0.
  - Strength: Default, 0.9, and 1.0.
  - Negative conditioning: $\emptyset$ (no negative prompt), and "No acute cardiopulmonary process."

  Default settings were used for all other underlying model hyperparameters.

**Implementation details.** *SemiSynCXR*'s editing component is built upon the Stable Diffusion Inpainting pipeline from the HuggingFace `Diffusers` library (von Platen et al., 2022). For the underlying models, RoentGen weights were provided by the authors (version dated December 31, 2023), while RadEdit weights were sourced directly from the HuggingFace Hub. To mitigate bottlenecks during semi-synthetic image generation, we set the numerical precision to `bfloat16`, which provides a favorable balance between memory efficiency and numerical stability. Experiments were conducted on an NVIDIA RTX A6000 GPU (48GB VRAM) and the compute cluster from the Chair of AI in Healthcare and Medicine, specifically its nodes equipped with NVIDIA A40 GPUs.

