# OpenReview forum: "Semi-Synthetic Localization Datasets for Radiological Findings on Chest X-Rays"
_MIDL.io/2026/Conference — MIDL 2026 Poster_

### Official Review · Reviewer_H2vc · 2025-12-30

**Confidence:** 4
**Preliminary Rating:** 4
**Final Rating:** 4

**Summary:**

SemiSynCXR generates semi-synthetic chest X-rays with localized radiological findings by inpainting into healthy CXRs using mask- and text-conditioned latent diffusion models, producing precise ground-truth bounding boxes. Supplementing real datasets improves detection accuracy and generalization. Quantitative and qualitative evaluations confirm realistic images, accurate localization, and strong utility for model training.

**Strengths:**

SemiSynCXR generates realistic, localized chest X-rays, provides precise bounding boxes, improves model training, addresses data scarcity, and is well-structured, novel, rigorously evaluated, and impactful.

**Weaknesses:**

Depends on pre-trained models; mask assumptions may reduce realism; some findings (e.g., pneumothorax) are hard to model; size/severity not captured; limited expert evaluation; potential dataset bias.

**Detailed Comments:**

The paper could benefit from including more diverse datasets to reduce potential bias and improve generalization across different populations and imaging conditions. Expanding the expert evaluation with a larger sample size would strengthen the validity of the qualitative assessment. Clarifying the limitations in modeling complex findings, such as pneumothorax, and discussing the impact of mask assumptions on realism would provide a more complete understanding of the method’s constraints. Additionally, guidance on the optimal ratio of synthetic to real data for training object detectors would help practitioners apply SemiSynCXR more effectively. Reporting computational costs and runtime would improve reproducibility, while adding more visual examples comparing real versus semi-synthetic CXRs would enhance interpretability. Finally, outlining future plans for extending the framework to additional radiological findings would demonstrate the method’s broader applicability and potential for continued development.

**Justification Of Final Rating:**

"  My initial recommendation of weak accept remains unchanged overall, as the rebuttal satisfactorily addresses and thoroughly resolves the major concerns raised during the review process effectively.  ”

**Justification Of The Preliminary Rating:**

Weak accept: Novel semi-synthetic CXR generation improves localization and training data. Clear method and results, but relies on pre-trained models, mask assumptions, limited evaluation, and some findings are hard to model.

**Questions To Address In The Rebuttal:**

I would like the authors to clarify the limitations of their mask generation assumptions and their impact on realism, especially for complex findings like pneumothorax. Details on how the framework handles variability in size and severity of findings, as well as justification for the optimal ratio of synthetic to real data for training, would strengthen the evaluation. Expanding on the expert assessment methodology, including sample size and selection criteria, would help assess the reliability of qualitative results. Additional discussion on potential dataset biases, generalization to diverse populations and imaging conditions, and computational requirements would also be valuable. Addressing these points could clarify the method’s practical applicability and limitations, which might influence my assessment of the paper’s overall impact and utility.

---

> ### Author Response · Authors · 2026-01-24
>
> We thank the reviewer for their detailed and beneficial feedback.
>
> 1. **Modeling complex findings and impact of mask assumptions on realism** [W2-3, D3, Q1]. We agree that modeling non-focal, diffuse medical conditions is challenging for our approach due to its bounding-box-constrained editing. While this editing approach is beneficial for maintaining structural integrity and provides precise ground truth localization annotations, it may not fully capture the nature of diffuse diseases. We have expanded the Discussion section (p. 12) to further acknowledge this.
> 2. **Variability in size and severity of findings** [W4, Q2]. Following the reviewer’s suggestion, we have updated the Discussion section (p. 12) to further clarify that explicit control over the size and severity of findings is not yet supported and remains a priority for future work. Currently, these attributes are instead influenced by the sampled editing mask.
> 3. **Optimal ratio of synthetic to real data for training** [D4, Q2]. We consider that a full ablation study on the optimal real-to-synthetic ratio is required before providing a definitive conclusion; such a study is currently beyond the scope of this work. However, we have now included a preliminary analysis of this question in the Results section (p. 9) to provide initial insights.
> 4. **Additional details about the expert assessment methodology and its potential extension** [W5, D2, Q3]. We appreciate the suggestion to expand the expert assessment. Our current expert assessment included 140 CXRs (70 real, 70 generated) randomly selected to mitigate selection bias. We have updated the Results section (p. 11) to specify the total population size from which these samples were drawn. Although a larger-scale study was not feasible within the rebuttal timeframe, we have identified this as future work in the Discussion section (p. 12) and underscored the benefit such an evaluation would bring to the framework.
> 5. **Potential dataset biases, generalization to diverse populations, and imaging conditions** [W6, D1, Q4]. We would like to underscore that our evaluation incorporates the VinDr-CXR dataset, which represents a distinct patient population and imaging protocol compared to  MIMIC-CXR. A paragraph to this effect has been added to the Methodology section (p. 5). Among the framework's components (i.e., healthy input CXRs, editing-conditioning elements, and the underlying model), we identified healthy input X-rays as the source of the highest potential bias. Therefore, we recognize that both the inclusion of more diverse data within the framework's components and validation across additional CXR datasets are essential to fully address demographic and site-specific disparities. Consequently, we have expanded our Discussion section (p. 12) to explicitly state this necessity.
> 6. **Computational requirements and additional examples** [D5, Q4]. To improve reproducibility, we have added an Implementation subsection (p. 8; Appendix E, p. 30), and the code will be released as stated. Additional semi-synthetic examples can be found in Appendix A.5 (pp. 22-24); these have been updated to include side-by-side comparisons with their corresponding real images for better interpretability.
> 7. **Dependence on pretrained models** [W1]. We contend that leveraging pretrained text-conditioned diffusion models is a strategic advantage of our approach, enabling us to benefit from advancements in the field without the need to train from scratch. However, to address the reviewer’s concern, we have added a clarification in the Discussion section (p. 12) noting that even if the underlying model must be trained from scratch, our framework remains viable since training requires only image-text pairs rather than manual bounding box annotations.
> 8. **Extension to model additional radiological findings** [D6]. We fully agree with the reviewer that outlining future plans to extend the framework to additional radiological findings would demonstrate the method’s broader applicability and potential for continued development. Accordingly, we have explicitly listed potential future findings (nodules, fractures, and calcifications) in the Discussion section (p. 12). For this extension, the framework primarily requires the probability distributions of the bounding boxes for each finding.
>
> *Abbreviations. W: Weaknesses, D: Detail Comments, Q: Questions To Address in The Rebuttal*

---

### Official Review · Reviewer_yYgV · 2026-01-09

**Confidence:** 4
**Preliminary Rating:** 4
**Final Rating:** 4

**Summary:**

The paper proposes a method to automatically generate semi-synthetic chest X-rays by inpainting specific radiological findings into real healthy CXRs. It creates images that are realistic and anatomically plausible while also providing ground-truth bounding boxes for each finding. Mask-conditioned inpainting with pre-trained diffusion models are used without retraining. Localization performance is improved in downstream tasks when using the generated images as additional training data.

**Strengths:**

To my knowledge, the mask-conditioned inpainting with pre-trained diffusion models without retraining is novel. It is also a practical and elegant design, which respects anatomical constraints and spatial distributions

Broad evaluation of quality and realism of the generated images, and ablation study, and convincing quantitative results (despite lack of statistical test).

Good discussion of the limitations and possible future directions.

**Weaknesses:**

There are no statistical tests to support claims such as “generated samples significantly improve performance.”

The method is demonstrated only on CXRs.

While practical, the dependence on pretrained models limits the method to domains where such models exist.

**Detailed Comments:**

“Comprehensive quantitative and qualitative evaluations confirm that …” Provide main results that show that (mAP, AUROC gains).

No statistical test to evaluate significance of performance differences in the training augmentation results. (In introduction, you even state “generated samples significantly improves …”)

Typos: “cureently”

The paper is anonymous (single-blind mentioned in the call)

**Justification Of Final Rating:**

The authors addressed my concerns by adding appropriate statistical testing. They also expanded the discussion on generalization and dependence on pretrained models. The work remains focused on CXRs and depends on availability of pretrained models, but these limitations are acknowledged and discussed. The method is novel and practical, the experiments are quite convincing.

**Justification Of The Preliminary Rating:**

Novel and practical method.
Experiments show improved localization performance when using these images as augmentation, supported by qualitative and quantitative analyses.
Minor limitations stated above, and the work is methodologically sound and offers useful contributions.

**Questions To Address In The Rebuttal:**

Provide main quantitative results in abstract.

Comment on possibility to extend the method to other modalities/diseases etc.

Discuss cases where a pre-trained diffusion model is unavailable.

---

> ### Author Response · Authors · 2026-01-24
>
> We thank the reviewer for their insightful feedback.
>
> 1. **Statistical significance of augmentation results** [W1]**.** We acknowledge that the use of the term “significantly” requires formal statistical validation. To evaluate the impact of data augmentation with SemiSynCXR, we performed a statistical analysis of the mAP differences between the top-performing augmented models (trained with 17.5k semi-synthetic samples for out-of-distribution testing and 35k for in-distribution testing) and the real-only baseline. We employed a Wilcoxon Signed-Rank test with the alternative hypothesis that the median of the differences is greater than zero ($H_1:\text{median}(\Delta \text{mAP}_{10:70})>0$). Pairwise samples were generated using fully random and stratified bootstrap resampling. Notably, for both $N=100$ and $N=1000$ iterations, the null hypothesis ($H_0:\text{median}\leq0$) was rejected at a confidence level of $\alpha=0.05$ across all scenarios (in-distribution and out-of-distribution testing). These results confirm that the performance gains are statistically significant and have been integrated into the Results section (pp. 8, 10).
>
> 2. **Generalization beyond CXRs** [W2, D1, Q2]**.** While this study focuses on chest X-rays (CXRs), we now discuss the potential extension of SemiSynCXR to other modalities and medical conditions. We have noted that SemiSynCXR is particularly well-suited for conditions with focal manifestations (e.g., tumors, fractures, aneurysms) due to its bounding-box-constrained editing mechanism. Additionally, the framework is potentially adaptable to other 2D and 3D imaging modalities, as well as time series of images. Such extensions would require an existing pretrained text-conditioned diffusion model (or data to train one) and the bounding-box probability distributions.
> 3. **Dependence on pretrained models** [W3, Q3]**.** We contend that leveraging pretrained text-conditioned diffusion models is a strategic advantage of our approach, enabling us to benefit from advancements in the field without the need to train from scratch. However, to address the reviewer’s concern, we have added a clarification in the Discussion section (p. 12) noting that even if the underlying model must be trained from scratch, our framework remains viable since training requires only image-text pairs rather than manual bounding box annotations.
>
> We have also integrated the main quantitative findings into the Abstract [D1, Q1] and corrected the identified typo [D3].
>
> *Abbreviations. W: Weaknesses, D: Detail Comments, Q: Questions To Address in The Rebuttal*

---

> > ### Comment · Reviewer_yYgV · 2026-01-25
> >
> > Thank you for the response, clarifications, and updated paper with statistical tests and discussions. I do not have further comments, I think it is a good paper for MIDL.

---

### Official Review · Reviewer_yZuV · 2026-01-09

**Confidence:** 4
**Preliminary Rating:** 4
**Final Rating:** 5

**Summary:**

This paper addresses an important challenge in medical image analysis: that lesion localizations are often scarce and expensive to obtain. The authors propose a semi-synthetic chest X-ray generation framework that edits real healthy CXRs by inpainting specific radiological findings into anatomically plausible locations. Because the controlling mask is known, the method can directly produce a ground-truth bounding box from it.

**Strengths:**

(1) This paper addresses an important practical problem in chest X-ray localization, where high-quality disease location annotations are scarce. It presents a promising direction showing that semi-synthetic data can help mitigate this limitation.

(2) The proposed approach leverages existing CXR diffusion models (e.g., RadEdit and RoentGen) rather than training new ones.

(3) Experimental results show that the generated CXRs can be effectively combined with real data to improve the performance of finding localization. Both quantitative and qualitative evaluations suggest that the synthesized CXRs contain the intended findings and are reasonably realistic.

(4) This manuscript is well-structured and well-written. The proposed framework is clearly explained, and the experimental setup is designed to support a good evaluation.

**Weaknesses:**

(1) Since the generated findings are constrained by the inpainting mask. This can limit performance for findings such as pneumothorax, where the abnormality may not be well localized to a fixed or compact region.

(2) The current framework does not provide explicit control over finding size or severity. As a result, the synthesized images may overrepresent prominent, easy-to-detect findings while underrepresenting subtle or small abnormalities. This could reduce the utility of the synthetic data for downstream tasks that require sensitivity to mild cases.

(3) Table 1 reports only the overall average performance, without disease-wise results. Providing per-finding results would help readers understand which findings benefit most from the synthetic augmentation and where the approach is less effective.

**Detailed Comments:**

see above

**Justification Of Final Rating:**

This paper addressed an important question that fit into the scope of MIDL. With the new experiments and more comprehensive discussions. I believe my major concerns are fully addressed, and I consider this a strong paper; I recommend it for acceptance.

**Justification Of The Preliminary Rating:**

This paper tackles an important problem with a promising approach. The experimental results provide solid support and highlight the potential of the proposed method. Reporting per-finding localization performance would further strengthen the work.

**Questions To Address In The Rebuttal:**

I suggest the authors report per-finding localization performance in addition to the overall average. This would help readers understand which findings benefit most and which benefit less and better guide practitioners on how to prioritize classes when applying the SemiSynCXR framework.

---

> ### Author Response · Authors · 2026-01-24
>
> We thank the reviewer for their constructive feedback. Addressing the question on **per-finding localization performance**, we have
>
> - Included Table 2 (Results, p. 9) and Table 4 (Appendix A.1, p. 18), providing per-finding AP$_{10:70}$ for the MS-CXR and VinDr-CXR test sets.
> - Added an analysis of these results in the Results section (p. 10).
>
> Our results indicate that the benefit of data augmentation with SemiSynCXR-generated data varies across findings. Specifically, findings that are difficult to automatically localize (e.g., pneumothorax) benefited most, whereas those with already high accuracy (e.g., cardiomegaly) saw more modest gains. Performance improved across all findings, except for atelectasis.
>
> Furthermore, we appreciate the reviewer pointing out two limitations of our approach: **(i) generated radiological findings are constrained by the editing mask**, and **(ii) our framework offers only non-explicit control over finding size and severity**. While these were briefly mentioned in our initial submission, we have expanded the Discussion section (p. 12) to further acknowledge these points and outline potential future strategies to address them.

---

> ### Comment · Reviewer_yZuV · 2026-02-02
>
> Thank you for the author's new experimental results and response to my concerns. I would recommend acceptance of this manuscript.

---

### Author Rebuttal · Authors · 2026-01-24

**Rebuttal:**

We thank the reviewers for their valuable feedback, which has substantially helped us improve the quality and clarity of our manuscript. Specifically, we have made the following major revisions:

- **Expanded discussion and scope.** We have updated the Discussion section to emphasize SemiSynCXR's potential for extension to other radiological findings, medical conditions, and imaging modalities. We now also extensively discuss the modeling of non-focal medical conditions, the current lack of explicit control over finding severity and size, mitigation of potential dataset biases, and the framework's ability to leverage existing pretrained models.
- **Per-finding data augmentation results and statistical significance.** We have added and analyzed the per-finding results for SemiSynCXR-generated images when used as data augmentation for finding localization (Tables 2 and 4). Furthermore, we have performed a Wilcoxon Signed-Rank test on the mAP$_{10:70}$ using bootstrap resampling ($N=100,1000$), confirming that overall gains are statistically significant ($p<0.05$) for both in-distribution and out-of-distribution testing.
- **Implementation details and improved graphics.** We have included details regarding the implementation and a preliminary analysis of the optimal ratio of synthetic-to-real data for localization training. Additional semi-synthetic examples, found in Appendix A.5, have been updated to include side-by-side comparisons with their corresponding real images for better interpretability.

**Supporting Material:**

/attachment/1c993737461459b07e540b5ad8e6797ebc4a5364.pdf

---

### Comment · Area_Chair_jLEs · 2026-01-29
**Discussion phase**

Dear Reviewers, the manuscript has now entered the discussion phase.

We kindly invite you to evaluate the authors’ responses and the revised manuscript, and to engage in discussion with the authors to address any remaining questions or unresolved points.

Once you have completed your review and discussion, submit your final rating. Please complete this step by selecting “Edit” → “Official Review” no later than February 1, 2026, at 23:59 AoE.

---

### Meta-Review · Area_Chair_jLEs · 2026-02-07

**Recommendation:** Accept (Oral)
**Confidence:** 5

**Metareview:**

This paper is a clear acceptance. Reviewers were unanimous in recognizing the utility of the proposed framework for generating semi-synthetic localization chest X-ray datasets. Furthermore, the evaluation was deemed robust, and the authors successfully resolved major concerns during the rebuttal phase.

---

### Decision · Program_Chairs · 2026-02-13

Accept (Poster)